

# A biofilter for treating toluene vapors: performance evaluation and microbial counts behavior

Yazhong Zhu, Shunyi Li, Yimeng Luo, Hongye Ma and Yan Wang

School of Chemical Engineering and Energy, Zhengzhou University, Zhengzhou, China

## ABSTRACT

A lab-scale biofilter packed with mixed packing materials was used for degradation of toluene. Different empty bed residence times, 148.3, 74.2 and 49.4 s, were tested for inlet concentration ranging from 0.2 to 1.2 $g/m^3$. The maximum elimination capacity of 36.0 $g/(m^3\ h)$ occurred at an inlet loading rate of 45.9 $g/(m^3\ h)$. The contribution of the lower layer was higher than other layers and always had the highest elimination capacity. The carbon dioxide production rate and distribution of micro-organisms followed toluene elimination capacities. The results of this study indicated that mixed packing materials could be considered as a potential biofilter carrier, with low pressure drop (less than 84.9 Pa/m), for treating air streams containing VOCs.

## INTRODUCTION

Large quantities of volatile organic compounds (VOCs) are emitted into the atmosphere from different resources, such as chemical, petrochemical, pharmaceutical, food processing, pulp and paper mills, color printing, painting works, vehicle exhaust, waste incinerators and composting facilities (*Chen, Fang & Shu, 2005*; *Slominska, Krol & Namiesnik, 2013*; *Yassaa et al., 2006*). Toluene is one of the common air pollutants in different industries. It is mutagenic and carcinogenic, and exposure to toluene might cause damage to the liver, kidney and the central nervous system (*Gallastegui et al., 2011*; *Rene, Murthy & Swaminathan, 2005*). According to the report of operating facilities in 2009, the rate of toluene emission into the atmosphere was 12.2 kt/yr in the USA, and 3.9 kt/yr in Canada (*Gallastegui et al., 2011*).

Biofilters are widely used for odor and air pollution treatment, particularly for VOCs with high flow rates and pollutants concentration less than 1,000 ppm (*Delhoménie et al., 2003a*; *Delhomenie et al., 2003b*; *El-Naas, Acio & El Telib, 2014*; *Maestre et al., 2007*; *Rahul, Mathur & Balomajumder, 2013a*; *Rahul, Mathur & Balomajumder, 2013b*; *Rene, Murthy & Swaminathan, 2009*; *Singh et al., 2010*). Compared to conventional technology, biofliters are cost competitive, with no secondary pollutants produced (*Elmrini et al., 2004*). Many references show that toluene could be used as a biofilter's substrate (*Aly Hassan & Sorial, 2009*; *Cho et al., 2009*; *Xi, Hu & Qian, 2006*).

Corresponding author
Shunyi Li, lsy76@zzu.edu.cn

**Table 1  Physical properties of the mixed packing materials.**

| Parameter | Units | Mixed packing materials |
|---|---|---|
| Equivalent diameter | mm | 10–12 |
| Bulk density | kg/m$^3$ | 471.0 ± 0.8 |
| Specific surface area | m$^2$/g | 3.91 ± 0.20 |
| Void space volume | % | 38–41 |
| Water holding capacity | % | 52 |

Packing materials are where physical, chemical and biological reactions occur; thus, the properties are concerned, such as high surface area and porosity for biofilm growth, suitable pH, acceptable buffering capacity (*Mudliar et al., 2010*; *Zare et al., 2012*) and benign water-holding capacity (*Anet et al., 2013*). Peat, soil, compost, barks and wood chips are the commonly used organic medias (*Lebrero et al., 2014*). Lifespans of such organic medias are short, and may cause clogging in the long run (*Dorado et al., 2010*). Other media such as perlite, vermiculite, glass beads, polyurethane foam, polystyrene and lava rock, may have indigenous micro-organisms and need extra nutrients (*Mudliar et al., 2010*).

*Singh, Rai & Upadhyay (2010)* evaluated the performance of a biofilter treating toluene packed with polyurethane foam. The removal efficiency ranged from 68.2 to 99.9% and elimination capacity ranged from 10.85 to 90.48 g/(m$^3$ h). The removal efficiency ranged from 40 to 95% and elimination capacity ranged from 3.5 to 128 g/(m$^3$ h) was observed by (*Rene, Murthy & Swaminathan, 2005*). However, few researchers focused on how the behaviors of different layers contributed to the overall performance, and if the relation between the microbial counts and the inlet loading rate were clear.

The main objective of this research was to determine the removal efficiency and elimination capacity of different layers as a function of inlet loading rate and empty bed residence time in a lab scale biofilter. The production of carbon dioxide and the microbial counts of three layers were also evaluated, and the variation of the pressure drops was observed.

## MATERIALS AND METHOD

### Inoculum and packing material

The inert material employed in the biofilter was invented by this lab (China invention patent, ZL201210446960.1), and was mixed by compost, cement, perlite, $CaCO_3$, plant fiber, etc. Sodium silicate was used as adhesive. The physical properties were summarized in Table 1. Fresh activated sludge was used as the inoculum source for the biofiter, which was obtained from a municipal wastewater treatment plant in Zhengzhou, China. Microorganisms in the activated sludge were acclimated to toluene in order to accelerate the adaptation period. For acclimation, one liter of the activated sludge was enclosed in an aerated tank and diluted with 3 L of nutrient solution (*Amin et al., 2014*). The composition of nutrient solution per liter of distilled water was: $K_2HPO_4$-0.11 g, $KH_2PO_4$-0.04 g, $NH_4Cl$-0.54 g, $MgSO_4$-0.067 g, $CaCl_2$-0.036 g, $FeCl_3$-0.25 mg, $MnSO_4$-0.03 mg, $ZnSO_4$-0.04 mg, $(NH_4)_6Mo_7O_{24} \cdot 4H_2O$-0.03 mg.

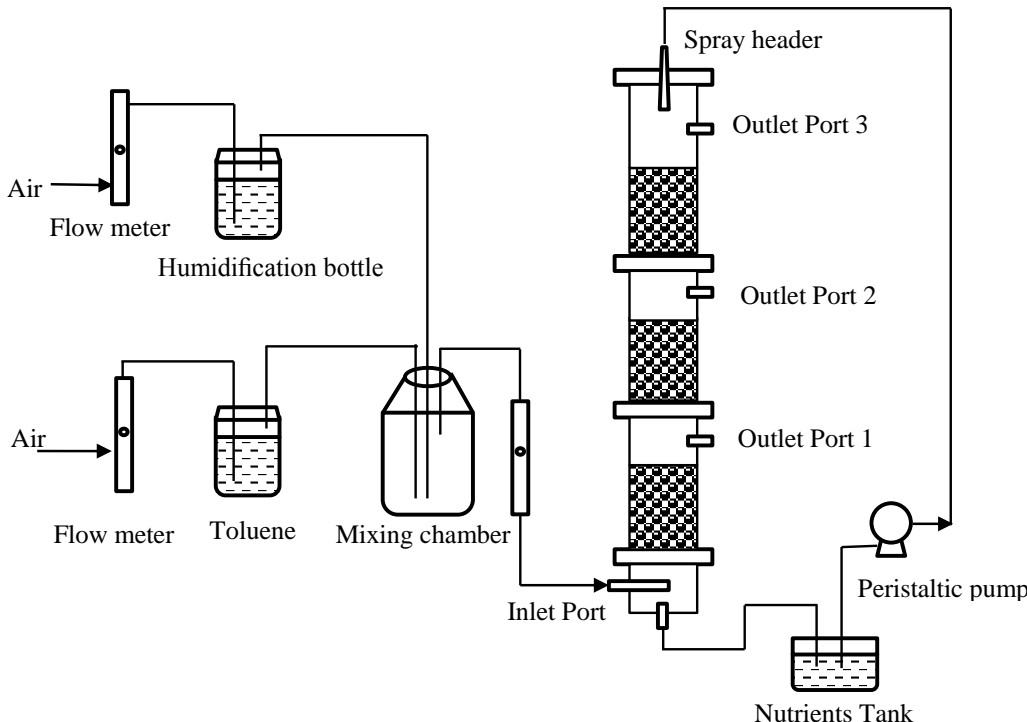

**Figure 1** Schematic diagram of the biofilter system.

## Biofilter setup and operation conditions

The biofilter was constructed from plexiglas cylinders with an internal diameter of 105 mm, and a total bed height of 90 mm, which was divided into three same sections. The total bed volume was approximately 8.24 L. Figure 1 shows the schematic diagram of the biofilter system. Toluene (99.5% AR Grade; Kemel, Shanghai, China) was stripped with compressed air. The biofilter was operated in an up-flow mode at room temperature. The concentration of pollutants was fixed by means of flowmeters (all from Yuyao Kingtai instrument Co., Zheijing, China).

The operating conditions of the biofilter are summarized in Table 2. During the study, different inlet loading rates (ILR), $5.0 \pm 1.0$, $15.2 \pm 1.8$, $25.6 \pm 2.9$, $34.4 \pm 2.0$, $44.5 \pm 1.5$ and $61.1 \pm 5.0$ g/(m$^3$ h), were set up at an empty bed residence time (EBRT) of 74.2 s. Experiments at EBRTs of 148.3 s and 49.4 s were also carried out, at ILRs of $24.4 \pm 2.9$ and $25.3 \pm 2.6$ g/(m$^3$ h), respectively. At each stage, inlet concentration of toluene was kept constant, and the biofilter was operated until pseudo steady-state when removal efficiency was constant. Microbial cell counts and carbon dioxide concentrations measured simultaneously. In order to insure satisfactory conditions of moisture and nutrients for microorganism activities, the nutrient solution was sprayed at a flow rate of 20 ml/min for 30 min every day, on the top of the packing media through the nutrient distribution system using a peristaltic pump.

**Table 2 Operating conditions of the biofilter.**

| Phase of operation | Gas flow rate (m³/h) | Inlet concentration (g/m³) | EBRT (s) | ILR (g/(m³ h)) | Operation times (days) |
|---|---|---|---|---|---|
| | | $0.10 \pm 0.02$ | | $5.0 \pm 1.0$ | 7 |
| | | $0.31 \pm 0.04$ | | $15.2 \pm 1.8$ | 7 |
| Phase I | 0.2 | $0.53 \pm 0.06$ | 74.2 | $25.6 \pm 2.9$ | 7 |
| | | $0.71 \pm 0.04$ | | $34.4 \pm 2.0$ | 7 |
| | | $0.92 \pm 0.03$ | | $44.5 \pm 1.5$ | 8 |
| | | $1.26 \pm 0.10$ | | $61.1 \pm 5.0$ | 10 |
| Phase II | 0.1 | $0.53 \pm 0.08$ | 148.3 | $24.4 \pm 2.9$ | 7 |
| | 0.4 | $0.36 \pm 0.05$ | 49.4 | $25.3 \pm 2.6$ | 10 |

## Analytical methods

Toluene concentration in the gas phase was measured using a gas chromatograph (GC1120; Sunny Hengping, China) equipped with a flame ionization detector (FID) and a FFAP chromatographic column (30 m × 0.25 mm × 0.25 μm; Nanjingjianuo, China). The nitrogen was used as a carrier gas at a flow rate of 0.4 ml/min. The oven, injector and FID detector was maintained at 65, 150 and 250°, respectively.

The pressure drop and temperature were measured by means of testo 510 and testo 405-V1 (Testo AG, Germany), respectively. The Moisture Content of packing materials was determined by the weight loss method after drying 12 h at 105 °C.

Carbon dioxide concentration in the gas phase was determined by the capacity titration method. $CO_2$ was first absorbed into $Ba(OH)_2$ solution (1.4 g/L), with an atmosphere sampler (QC-2B; Beijing Municipal Institute of Labor Protection, China). A 25 mL of the solution was titrated by $CH_3COOH$ solution (0.6 g/L), and phenolphthalein was used as indicator.

Microbial cell counts were measured by taken 1 g of moist media materials from three different locations at each layer of the biofilter. Each sample was mixed with 9 ml sterile extraction buffer (0.9% NaCl). The samples were subsequently shaken vigorously for 30 min, and serially diluted with sterilized water. Finally, 1 mL solution was plated in a nutrient agar for isolation of bacteria (*Rene, Murthy & Swaminathan, 2009*; *Saravanan & Rajamohan, 2009*). The composition of nutrient agar per liter was as follows: peptone-5 g, yeast extract-2.5 g, glucose-1.0 g and agar-15 g. The colonies were incubated for 3 days at 30 °C before counted.

## Performance evaluation

The parameters of the biofilter performance are illustrated in Table 3. The results are expressed in terms of inlet loading rate, elimination capacity and removal efficiency. Data from daily measurements were used to obtain average values of the biofilter.

# RESULTS AND DISCUSSION

## Influence of toluene inlet concentration

EC and RE of toluene as a function of ILR, during the Phase I, are illustrated in Fig. 2. ILR was gradually increased from 5.0 to 61.1 g/(m³ h). RE was almost constantly with the

**Table 3  Definition of biofilter performance parameters.**

| Parameter | Definition | Units |
|---|---|---|
| Empty bed residence time | $\text{EBRT} = \frac{V}{Q}$ | s |
| Inlet loading rate | $\text{ILR} = \frac{Q \times C_{in}}{V}$ | $g/(m^3\ h)$ |
| Elimination capacity | $\text{EC} = \frac{Q \times (C_{in} - C_{out})}{V}$ | $g/(m^3\ h)$ |
| Removal efficiency | $\text{RE} = \frac{C_{in} - C_{out}}{C_{in}} \times 100$ | % |
| Carbon dioxide production rate | $P_{CO_2} = \frac{Q \times (C_{out,CO_2} - C_{in,CO_2})}{V}$ | $g/(m^3\ h)$ |

**Notes.**

Where $Q$ is the total air flow rate ($m^3/h$); $V$ is the empty bed volume ($m^3$); $C_{in}$ and $C_{out}$ are the inlet and outlet concentration of toluene, respectively. $C_{in,CO_2}$ and $C_{out,CO_2}$ are the inlet and out concentration of carbon dioxide.

**Table 4  Comparison of biofilter performance.**

| References | Pollutants | Packing media | EBRT (s) | $EC_{max}$ $(g/(m^3\ h))$ | RE of $EC_{max}$ (%) | Micro-organisms |
|---|---|---|---|---|---|---|
| *Zamir, Halladj & Nasernejad (2011)* | Toluene | Compost and lava | 264 | 1.9 | 92 | Fungi |
| *Gallastegui et al. (2011)* | Toluene | Small stones | 180 | 40.3 | 69.6 | Bacteria |
|  | p-xylene |  |  | 26.5 | 40.0 |  |
| *Singh, Rai & Upadhyay (2006)* | Toluene | Agro waste | 154 | 174.6 | 59.8 | Activated sludge |
| This work | Toluene | Mixed media | 74.2 | 36.0 | 78.4 | Activated sludge |

increased of ILR up to 34.4 g/(m³ h); then it decreased. The corresponding EC was linearly increased with ILR from 5.0 to 34.4 g/(m³ h). Maximum EC was 36.0 g/(m³ h) occurred at an ILR of 45.9 g/(m³ h). After that the EC decreased, and RE was only 50.6% under an ILR of 61.1 g/(m³ h). Two distinct zones were observed in the RE versus ILR graph. The results obtained above were in agreement with *Singh et al. (2010)*, *Elmrini et al. (2004)* and *Kiared et al. (1997)*. Comparison of biofilter performance is given in Table 4.

Zamir and colleagues investigated a compost biofilter treating toluene vapor; maximum RE and EC was 92% and 1.9g/(m³ h), respectively. The $EC_{max}$ was far less than this study; this might be explained by the biofilter they used was dominated by the white-rot fungus. Gallstegui and colleagues evaluated biofiltration of toluene and p-xylene; $EC_{max}$ of 40.3 g/(m³ h) was observed. The reasons for their better performance could be that the longer operation of EBRT and the existence of p-xylene may stimulate the degradation of toluene. Singh and colleagues got a higher $EC_{max}$ of 174.6 g/(m³ h); however, the RE was much lower.

## Influence of gas flow rate

The gas flow rate is an important parameter in biofilter operation. Three levels of gas flow rate, i.e., 0.1, 0.2 and 0.4 m³/h, were performed. RE and EC as a function of EBRT are shown in Fig. 3. ILR of 148.3, 74.2 and 49.4s were set at the same levels, which were 24.4, 25.6 and 25.3g/(m³ h), respectively. Depending on Fig. 3, when EBRT decreased from 148.3 to 74.2 s, biofilter maintained high RE. However, when EBRT decreased to 49.4 s, RE decreased to 71.0%; this might be because reduction in the contaminant retention time could not provide sufficient time for toluene to transfer into biofilm. The results were

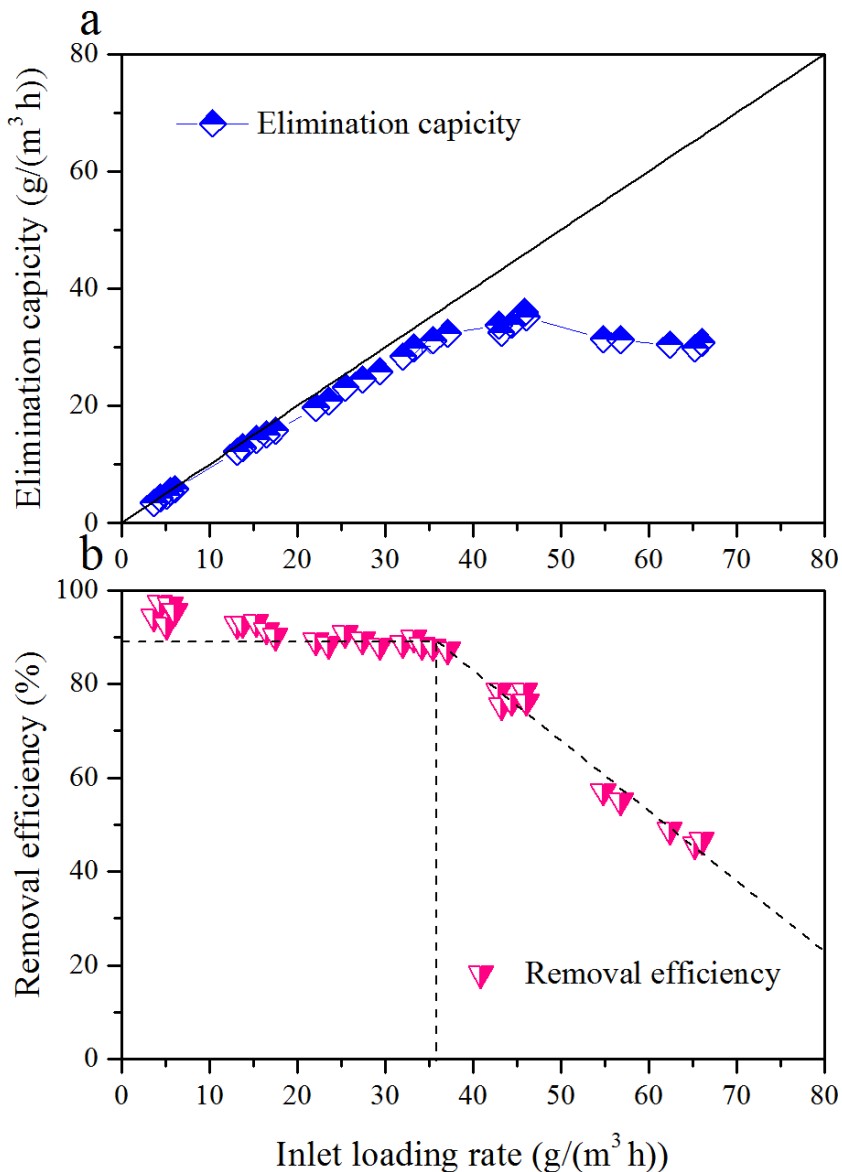

**Figure 2** Influence of inlet loading rate on the elimination capacity (A) and removal efficiency (B) of the biofilter at an EBRT of 74.2 s.

coordinated with the findings of some literature; biofilter performance decreased with decreasing EBRTs (*Abumaizar, Kocher & Smith, 1998*; *Rene et al., 2012*).

## Evaluation of $CO_2$

Toluene was finally biodegraded to $CO_2$ and $H_2O$, and utilized to format biomass for microbial growth (*Andreoni & Gianfreda, 2007*); thus, monitoring $CO_2$ concentration provided valuable information for the degree of VOCs mineralization. The stoichiometric reaction of toluene oxidation can be written as follows:

$$C_7H_8 + 9O_2 \rightarrow 7CO_2 + 4H_2O. \tag{1}$$

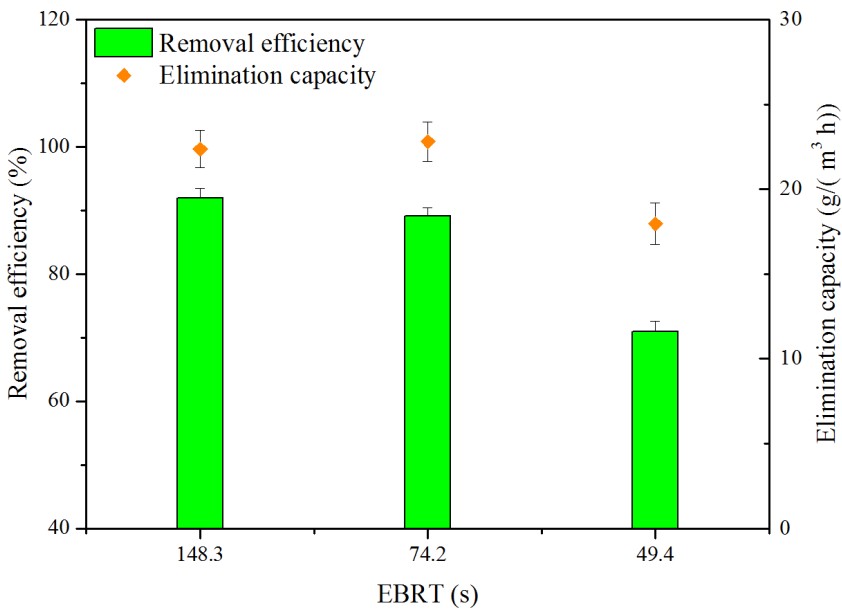

**Figure 3** Influence of EBRT on removal efficiency and elimination capacity.

$P_{CO_2}$ during Phase I as a function of EC for toluene is shown in Fig. 4. The $P_{CO_2}$ was concluded to linearly increase along with the EC at Phase I. A linear regression, calculated according to the least square method, provided the following equations for toluene degradation:

$$P_{CO_2} = 1.45EC - 1.23. \qquad (2)$$

The mass-ratio of $P_{CO_2}$ to EC of toluene was 1.45, less than the theoretical calculation. The theoretical mass-ratio should be 3.35, when the toluene was totally oxidation to $H_2O$ and $CO_2$. *Gallastegui et al. (2013)* reported the biodegradation of ethylbenzene and toluene. According to their study, linear fits to experimental data was made, and mass-ratios of ethylbenzene and toluene were 1.36 and 2.84, respectively. Cheng and colleagues (*2016*) reported biodegradation of toluene in fungal biofilter (F-BF), bacterial biofilters (B-BF) and fungal & bacterial biofilters (F & B-BF). The mass-ratio of F-BF, F & B-BF, and B-BF was 1.23, 2.52, and 2.85, respectively.

The cause of discrepancy might be that the biodegradation of toluene took some steps to convert into biomass or product $CO_2$, and some intermediates may not degrade immediately. In addition, some of the $CO_2$ could accumulate in the liquid in other forms, such as $CO_3^{2-}$, $HCO_3^-$ and $H_2CO_3$ (*Wu et al., 2006*).

## Evaluation of different layers

The biofilter was subdivided into three identical layers; gas samples were collected from each port of the biofilter. In order to have an insight into the contribution of different layer to its overall performance, the RE and EC of the three layers as a function of ILR is shown in Fig. 5.

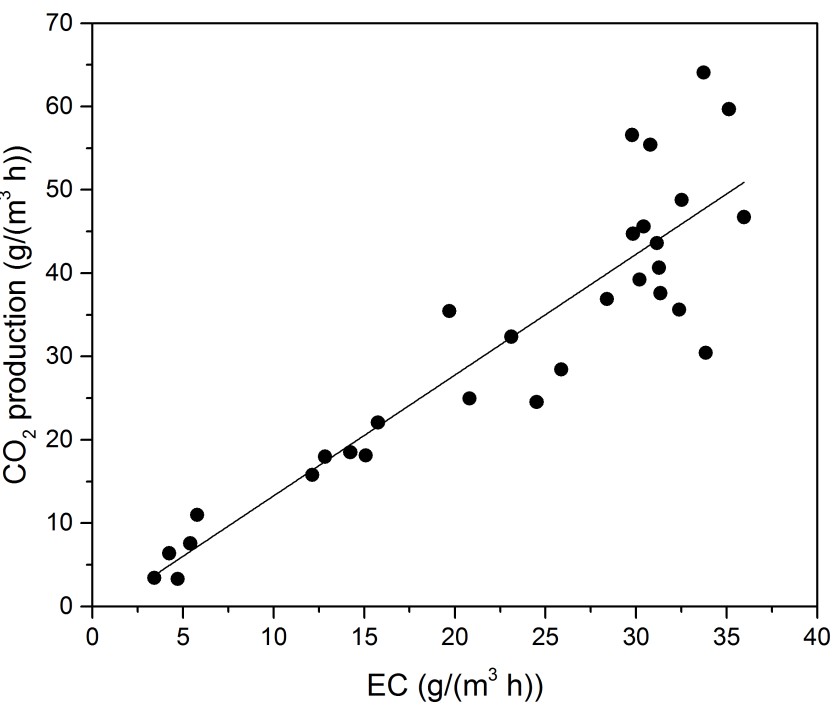

**Figure 4** Carbon dioxide production rate as a function of EC for toluene.

Results illustrated that contributions changed depending on ILRs. As ILR increased, the RE of the lower layer decreased from 85.1 to 21.5%; while the middle layer improved from 7.5 to 29.4 (at an ILR of 25.6 g/(m³ h)) then decreased to 16.2%; and the upper layer improved from 2.7 to 29.0 (at an ILR of 34.4 g/(m³ h)) then decreased to 12.9%. At low ILR, RE was mostly contributed by the lower layer. The majority of the toluene were eliminated in the lower layer; only a small portion of toluene was offered to the middle and upper layers. When at a higher ILR, the toluene cannot be completely degraded by the lower layer, and the rest flowed into other two layers. However, the EC of the lower layer was still higher than other layers. The cause of the EC in the lower layer was always the highest, may be due to the higher microbial population and nutrients.

Similar results were noted in other researches (*Elmrini et al., 2004*; *Vergara-Fernandez et al., 2007*). *Rene et al. (2015)* studied the performance of a biofilter treating benzene and toluene, in an up-flow mode (same with the study). However, the results showed that the elimination of toluene was mostly occurred at the topside of the biofiter, which was not confirmed by the results in this study. This may be due to the biofiter used in that study was first to treat benzene, whereas the biofilter in this study used was only to treat toluene.

The carbon dioxide production rate at the three layers as a function of EBTT is shown in Fig. 6. From Fig. 6, it was clear that the highest EBRT the highest carbon dioxide concentration, since the micro-organism at this moment could obtain large amounts of contaminants. The carbon dioxide generated by the lower layer preceded the other two layers at the three EBRTs, and this was in accordance with the lower layer had larger elimination capacity of the results got above.

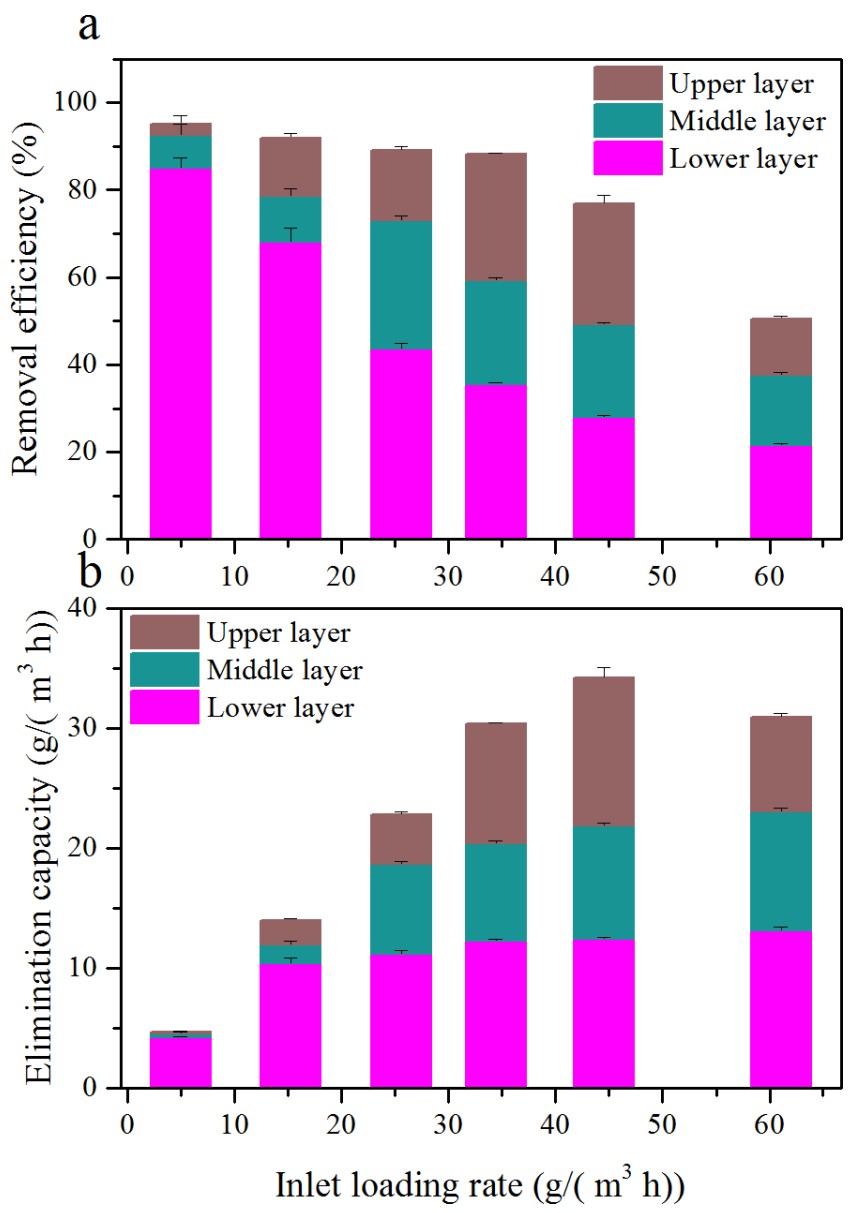

**Figure 5  Comparison of removal efficiency (A) and elimination capacity (B) among the three layers at various inlet loading rate.**

## Microbial counts behavior

According to the results of the microbial cell counts, there were mainly three kind of micro-organism, one kind of fungi and two kinds of bacteria. The fungus was white and filamentous, and the microbial count versus time is shown in Fig. 7. The microbial count of the two bacteria—one bacterium was pale yellow named bacterium-A, the other one was pinky named bacterium-B—versus time is shown in Fig. 8.

At the beginning of the operation, the microbial count of fungi at the three layers was at the same level, which was less than $10^3$ CFU/g. Then, it gradually increased to about $3.5 \times 10^5$ CFU/g at the lower layer, $3 \times 10^4$ CFU/g at middle layer, $6 \times 10^3$ CFU/g at the

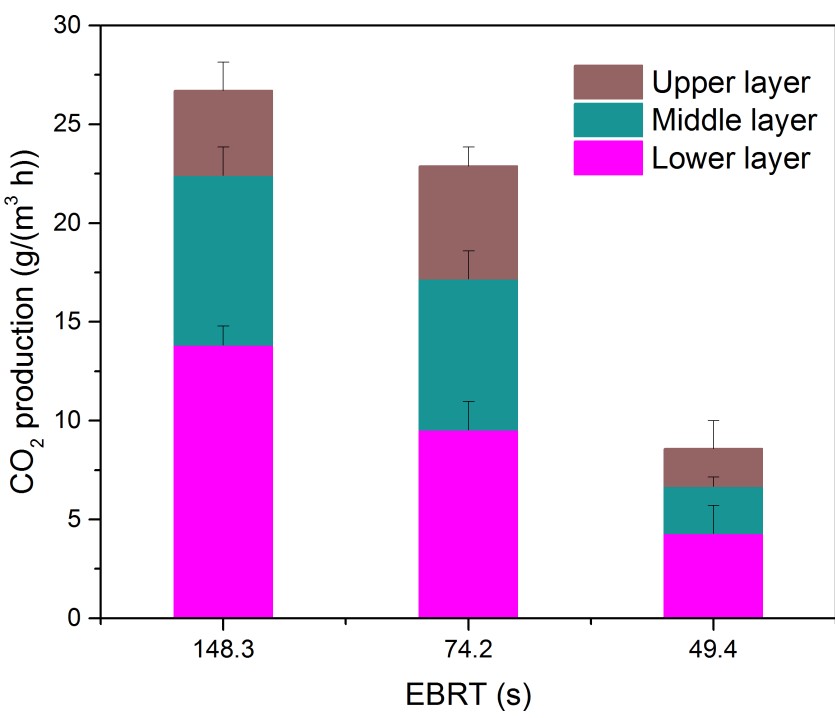

**Figure 6** Carbon dioxide production rate at the three layers as a function of EBRT.

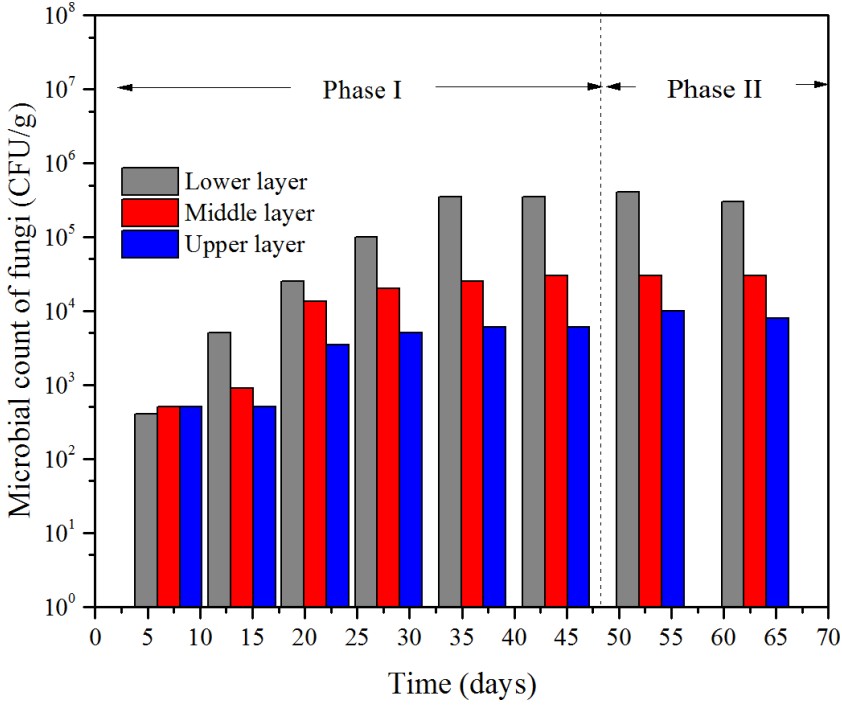

**Figure 7** Microbial count of fungi at the three layers of the bioflter versus time.

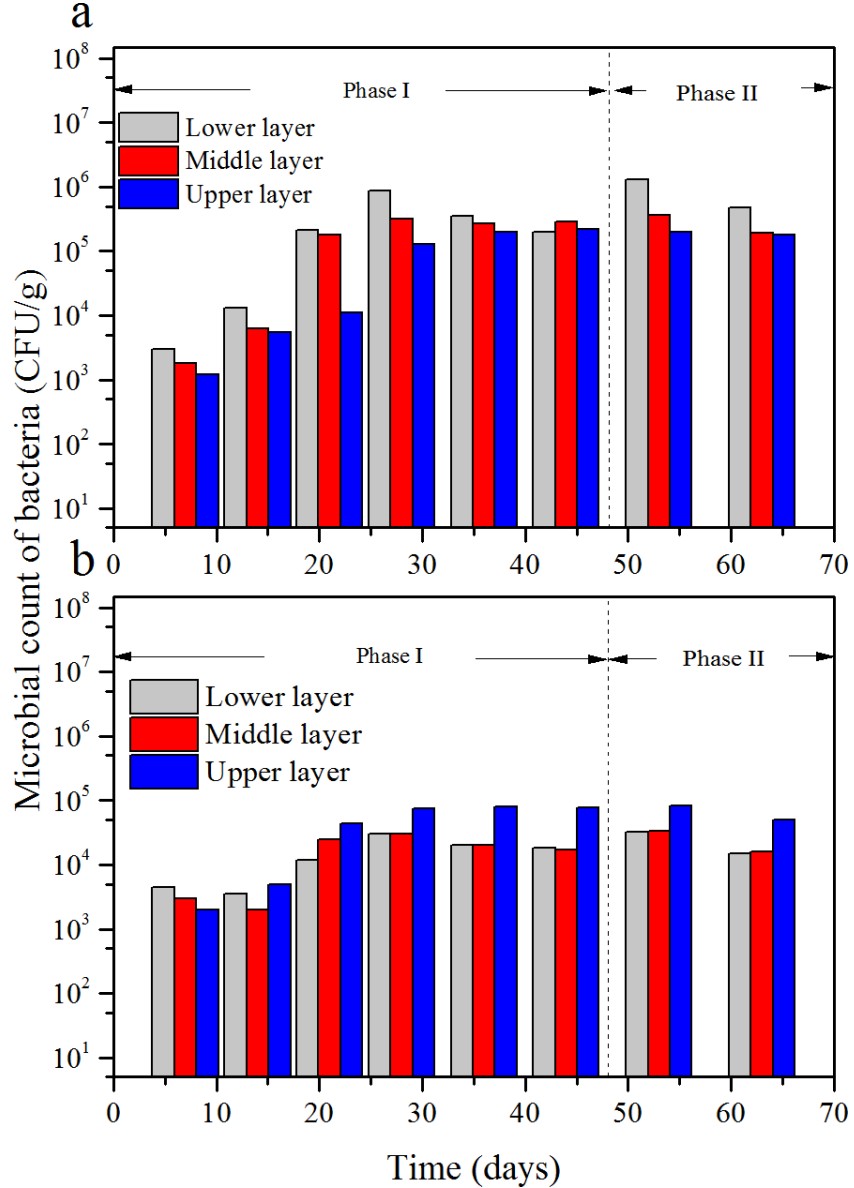

**Figure 8** Microbial counts of bacterium-A (A) and bacterium-B (B) at the three layers versus time.

upper layer, respectively, at the 36th day which the ILR was 44.5g/(m$^3$ h). The count of bacterium-A had the similar trend with fungi; however, the differences were the initial counts at the three layers were a little more than $10^4$ CFU/g and the maximum number occurred in the 28th day at an ILR of 34.4 g/(m$^3$ h).

However, when compared to bacterium-B, it showed some differences. Initially, count of bacterium-B was close to bacterium-A, $4.5 \times 10^3$ CFU/g in the lower layer, $3 \times 10^3$ CFU/g in the middle layer, $6 \times 10^2$ CFU/g in the upper layer, respectively. The count increased with the increase of the ILR, then maintained stability, which was similar with fungi and bacterium-A. However, for most of the time the microbial count of the upper layer was

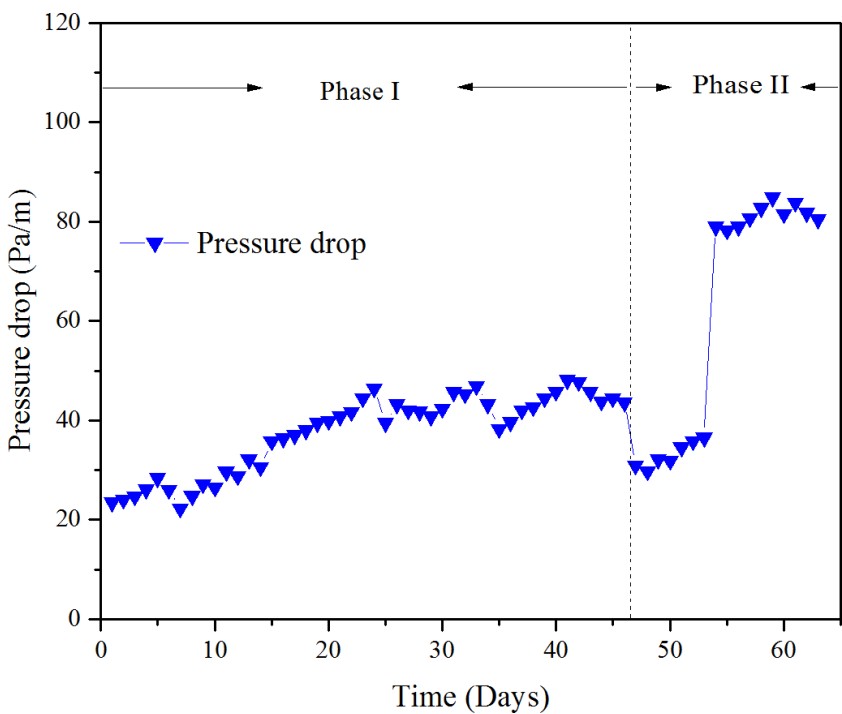

**Figure 9** Pressure drop versus time at various phase.

higher than that of the other two layers, and a maximum value of $8 \times 10^2$ CFU/g was achieved at an ILR of 44.5 g/(m³ h).

Both the microbial counts of the fungi and the bacteria were depended on ILR, which demonstrated that the micro-organisms were fed on the contaminants. The trend of the micro-organisms at different layers under various ILRs was consistent with the trend of RE and EC. According to *Gallastegui et al. (2013)*, the microbial population and reaction capacity remained low at the lower layer, this was consistent with bacterium-B, however, but was not consistent with the trends of the fungi and bacterium-A. In their study, the concentration of the contaminant could achieve to 8.72 g/m³, because that the lower layer had the highest microbial population. The reason bacterium-B was higher at the upper layer may be that it was more sensitive to the concentration of the contaminant. The results of *Saravanan & Rajamohan (2009)* showed that the removals were more efficient in the lower layer which was consistent with the results got here.

## Behavior of the pressure drop

Pressure drop of the biofilter depends on many factors. The gas flow rate directly decided the velocity of the gas; the bigger the gas flow rate, the higher the pressure drop. Second was the media properties which include media size, porosity, depth and moisture content (*Singh, Agnihotri & Upadhyay, 2006*), Besides, the biomass accumulation in the biofilter may lead to changes in media bed characteristics, which may cause channel diminished, thus increased pressure drop (*Morgan-Sagastume, Sleep & Allen, 2001*). The pressure drop versus time is shown in Fig. 9.

The initial pressure drop during phase I was about 20 Pa/m, then increased slowly with the operation time, and finally achieved a steady state about 43 Pa/m. During phase II, the pressure drop decreased to nearly 30 Pa/m with the doubled EBRT, then increased to 81 Pa/m at an EBRT of 49.4 s; the sudden increase of pressure drop was due to reduction of EBRT. During phase I, the gas flow rate was maintained constant; thus, the increase of the pressure drops was mainly due to biomass accumulation. In addition, the bed compaction and deterioration was observed negligible, which indicated the mixed packing material had a good mechanical strength. The maximum value of the pressure drops was 84.9 Pa/m, which was significantly advanced to some organic materials for wood chips with a pressure drop of 2,600 Pa/m (*Morgan-Sagastume, Sleep & Allen, 2001*), and matured compost with a pressure drop of 264.8 Pa/m (*Delhoménie et al., 2003a*; *Delhomenie et al., 2003b*).

## CONCLUSION

In this paper, toluene was treated with an up-flow lab scale biofilter filled with inert packing materials. The $EC_{max}$ was observed at an inlet loading rate of 45.9 g/(m$^3$ h), and two distinct zones were also observed. During the whole operation, the highest EC appeared at the lower layer. The $CO_2$ production rate and the distribution of microbial populations in the biofilter were well correlated with the toluene removal efficiencies and elimination capacities, indicating the biodegradation of toluene in the biofilter. The low pressure drop demonstrated that the packing materials were proper for biofiltration.

### Funding

Financial support was provided by the National Natural Science Foundation of China (NO.U1304216), the Research Fund of Yong Scholars for the Doctoral Program of Higher Education of China (Grant No. 20124101120015), and the Foundation of He'nan Educational Committee (Grant No. 13A610689). The funders had no role in study design, data collection and analysis, decision to publish, or preparation of the manuscript.

### Grant Disclosures

The following grant information was disclosed by the authors:
National Natural Science Foundation of China: U1304216.
Research Fund of Yong Scholars for the Doctoral Program of Higher Education of China: 20124101120015.
Foundation of He'nan Educational Committee: 13A610689.

### Competing Interests

The authors make use of (and report on the performance of) a patent which was invented by our lab at the School of Chemical Engineering and Energy, Zhengzhou University, China.

## Author Contributions

- Yazhong Zhu conceived and designed the experiments, performed the experiments, analyzed the data, contributed reagents/materials/analysis tools, wrote the paper, prepared figures and/or tables, reviewed drafts of the paper.
- Shunyi Li conceived and designed the experiments, reviewed drafts of the paper.
- Yimeng Luo and Hongye Ma reviewed drafts of the paper.
- Yan Wang contributed reagents/materials/analysis tools.

## Patent Disclosures

The following patent dependencies were disclosed by the authors:

Patent: Functional microbe filling material embedded with slow-release composite and its preparation method; Owners: Shunyi LI, Rencheng Zhu, Yan Wang, Yali Zhang; ZL201210446960.1.

## Data Availability

The raw data has been supplied as Data S1.

## Supplemental Information

Supplemental information for this article can be found online at http://dx.doi.org/10.7717/peerj.2045#supplemental-information.

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
