# Peer review of "A biofilter for treating toluene vapors: performance evaluation and microbial counts behavior"

_PeerJ, doi:10.7717/peerj.2045_

## Round 0.1 · original submission · Major Revisions

The reviewers have provided important suggestions for improvement of the manuscript. Please provide a point by point rebuttal to the reviewers comments when submitting a revised version of the manuscript. An important point raised by both reviewers is that the obtained results should be compared with literature results on biofilters for toluene removal.

Please consider having the manuscript reviewed by an English language expert, because the text contains many language errors. Furthermore the numerical data presented contain too many digits. Use as a rule of thumb that numerical results should be rounded off such that the change introduced by rounding should be smaller than 10% of the standard error. So for example 34.42 +/- 1.97 should be changed into 34.4 +/- 2.0, 44.51 +/- 1.46 into 44.5 +/- 1.5 and so on.

·

Basic reporting

Lot of work on biofiltration of toluene has been done by various researchers and this work is quite similar to previous reported studies. The only interesting points I found in this paper is detail study of behavior of different layers of biofilter column and microbial population in different layers.

Experimental design

The experiments are well -planned.

Validity of the findings

Finding of the present study is publishable but I think the author should include a comparative table to compare the results of toluene degradation in the present study with previous studies.

Additional comments

As lot of studies are already available on this topic, the authors may easily shorten the manuscript by deleting the portions which are very common. The introduction and result and discussion portion may be reduce keeping the important facts intact.

Reviewer 2 ·

Basic reporting

 Language can be improved
 Some comment about the intro (see questions added). Literature references OK
 Structure Ok
 Figures … OK
 Raw data OK

Experimental design

 Scope OK.
 Original not in the sense that new scientific insight is presented. This is quite routine type of experiment in the field of biofiltration. The novelty is the material used which is patented. In this way it is more a prove of concept of the bio filter material.
 Research question is defined.
 Method description can be improved (see questions)

Validity of the findings

 In my opinion the results are too much presented as standalone results. I think that there is a lot more information in the literature about e.g. toluene degradation capacity in biofilter. The paper can be improved by comparing and discussing the authors results with literature data.
 There are some repetitions (see questions)

Additional comments

Abstract
1. Present information in a generic way. As independent of the experimental set up as possible; E.g. EBRT is generic flow rate is not generic. So change m3/h for EBRT
2. Numerical information should reflect the accuracy and reproducibility of the measurement. E.g. in the paper all data are given with 2 digits behind the decimal dot. But an EC of 61.07 g/m3h not be measured. So reporting the .07 makes no sense here.
Introduction
Ln 25 “highly toxic” this is a subjective statement. Compared with benzene, toluene is less toxic.
Ln. 37-38. Henry’s law coefficient does not impact biodegradations. Henry’s law impact determines equilibrium partitioning between air and water. So it impacts the liquid phase concentration but not biodegradation as such.
Ln 51 3inorganic materials …” then a list of materials follows but several materials (bark, polyurethane …) are not inorganic
Ln 74 sentence not clear
Ln 93 the first time an abbreviation is used it should be written full out as well
Ln 97 What is the meaning of “relatively constant” please give a standard deviation on the measured inlet concentrations.
Ln 101-102 “continuously sprayed every day” strange phrase
Ln 111-113 the carrier gas flow rate reported seems very high for a 0.25 mm diameter column; please check
Ln 119 The information given on microbial cell counts is not specific. Which kind of culture media was used? Later results for bacteria and fungi counts are given. Were these species grown on the same medium?
Ln 151 “Figure 1.” Seems to be an error. Figure 1 is a diagram of the experimental set up.
Ln 151-154 “When at relatively …” strange sentence. A range of ILR is given and then 4 values for RE. Where does those 4 values stand for?
Ln 164 “… pollutants …” There is only one pollutant toluene.
Ln 173 layers instead of layer
Ln 163-168 this is common knowledge; this paragraph could be omitted.
Ln 190-197 is a repetition of the mass transfer and bioreaction control operation; could be omitted.
Ln 198 was the biofilter used by Rene (2015) operated in up flow or in down flow mode?
LN 224 Could the authors make a mass balance between toluene degrade and CO2 produced? This would increase the value of including CO2 data
Figure 8. the pressure drop curves shows a sudden increase at day 53 or so. What is the explanation?

---

## Round 0.2 · accepted · Accept

All comments and suggestions for improvement raised by the reviewers have been addressed in the revised version of the manuscript.